# Joint Detection and Classification of Singing Voice Melody Using Convolutional Recurrent Neural Networks

**Sangeun Kum** and **Juhan Nam** *

Music and Audio Computing Lab, Graduate School of Culture Technology, KAIST, 291 Daehak-ro, Yuseong-gu, Daejeon 34141, Korea; keums@kaist.ac.kr
* Correspondence: juhannam@kaist.ac.kr; Tel.: +82-42-350-2926

**Abstract:** Singing melody extraction essentially involves two tasks: one is detecting the activity of a singing voice in polyphonic music, and the other is estimating the pitch of a singing voice in the detected voiced segments. In this paper, we present a joint detection and classification (JDC) network that conducts the singing voice detection and the pitch estimation simultaneously. The JDC network is composed of the main network that predicts the pitch contours of the singing melody and an auxiliary network that facilitates the detection of the singing voice. The main network is built with a convolutional recurrent neural network with residual connections and predicts pitch labels that cover the vocal range with a high resolution, as well as non-voice status. The auxiliary network is trained to detect the singing voice using multi-level features shared from the main network. The two optimization processes are tied with a joint melody loss function. We evaluate the proposed model on multiple melody extraction and vocal detection datasets, including cross-dataset evaluation. The experiments demonstrate how the auxiliary network and the joint melody loss function improve the melody extraction performance. Furthermore, the results show that our method outperforms state-of-the-art algorithms on the datasets.

**Keywords:** melody extraction; singing voice detection; joint detection and classification; convolutional recurrent neural network

---

## 1. Introduction

Melody extraction is estimating the fundamental frequency or pitch corresponding to the melody source. In popular music, the singing voice is commonly the main melodic source; thus, extracting the pitch of the singing voice in polyphonic music is the most common task. The extracted melody is useful in many ways. For example, the result can be directly applied to melody-based retrieval tasks such as query-by-humming [1] or cover song identification [2]. The pitch contour that contains the unique expressiveness of individual songs can be used as a feature for high-level tasks such as different music genre classification [3] or singer identification [4].

Singing voices are usually mixed to be louder than background music played with musical instruments in music production [5]. Furthermore, singing voices generally have different characteristics from those of music instruments; they have expressive vibrato and various formant patterns unique to vocal singing. A number of previous methods exploit the dominant and unique spectral patterns for melody extraction leveraging prior knowledge and heuristics. For example, they include calculating the pitch salience [6–10] or separating the melody source [11–13] to estimate the fundamental frequencies

of melody. In contrast, a data-driven approach using machine learning [14] has also been proposed. Recently, deep learning has been the main approach, as it has proven to be very successful in a wide variety of fields. Researchers have attempted various deep neural network architectures for melody extraction. Examples include fully-connected neural networks (FNN) [15,16], convolutional neural networks (CNN) [17,18], recurrent neural networks (RNN) [19], convolutional recurrent neural networks (CRNN) [20], and encoder-decoder [21,22].

Singing melody extraction involves detecting voice segments because the melodic source is not always active in the music track. Voice detection is a very important task that affects the performance of melody extraction. The methods for singing voice detection can be roughly divided into three approaches. One approach is detecting the voice segments by thresholding the likelihood of pitch estimation [15,17,18]. This method is very simple, but more inaccurate because the optimal threshold value may vary depending on the level ratio between the voice and background sound. Another approach is constructing a separate model for singing voice detection [10,16]. Since the model is dedicated to the voice detection, it can achieve better performance. However, it is obvious that it takes more effort to train the model separately, and the complexity of the whole melody extraction algorithm increases. The last approach is adding an explicit "non-voice" label to a list of target pitch outputs. This is particularly valid in classification-based melody extraction methods [19,20,23]. When singing voice is present, one of the target pitches is chosen as an output class. Otherwise, the "non-voice" label is chosen as another output class. This "none of the above" labeling method is also known to have a regularization effect [24].

Our proposed melody extraction model is based on the last approach. However, we pay attention to the discrepancy of the abstraction level between voice detection and pitch classification. Voice detection is mainly based on vibrato or formant modulation patterns that distinguish it from other musical instruments. This requires a much wider context than identifying pitch from voice segments, which can be estimated even from a single frame of audio. That is, the voice/non-voice discrimination is a higher-level task than the pitch classification. We handle this dual-target problem using a joint detection and classification (JDC) network. JDC can be regarded as multi-task learning [25]. It has been shown to improve generalization by combining task-specific information contained in network parameters for each related task. In the area of music or audio, it has been applied to tasks such as source separation [26,27] and bird audio detection [28]. In both tasks, detecting the source of interest, that is vocal or bird sound, is an important subtask that affects the performance of the main task. An additional network was built to detect vocal or bird sound and was integrated with the main network to improve the performance. Likewise, we also added an auxiliary network (*AUX*) trained to detect singing voice on top of the main melody extraction network.

In summary, the contributions of this paper are as follow: first, we propose a CRNN architecture with residual connections and bi-directional long shot-term memory (Bi-LSTM) as the main network so that the model classifies the pitch with a high resolution. Second, beyond the melody classification model, we propose a JDC network that can perform two tasks in melody extraction (singing voice detection and pitch classification) independently. The auxiliary network that detects the singing voice is trained using multi-level features shared from the main network. Third, we propose a joint melody loss designed to combine the two tasks together for the JDC network and to be effective for generalization of the model. Finally, by comparing the results on public test sets, we show that the proposed method outperforms state-of-the-art algorithms. Code and the pre-trained model used in this paper are available at https://github.com/keums/melodyExtraction_JDC.

## 2. Proposed Method

Figure 1 illustrates the overall architecture of the proposed singing melody extraction model. This section describes the details.

## 2.1. The Main Network

### 2.1.1. Architecture

The main network is the central part that extracts singing melody from polyphonic music audio. The architecture was built with 1 ConvBlock, 3 ResBlocks, 1 PoolBlock, and a bi-directional long short-term memory (Bi-LSTM) layer. The diagram of the architecture is shown in Figure 1a. The parameters and output sizes of the modules are listed in Table 1. ConvBlock is a module consisting of two $3 \times 3$ convolutional (Conv) layers, with a batch normalization (BN) layer [29] and a leaky rectified linear unit (LReLU) between them [30].

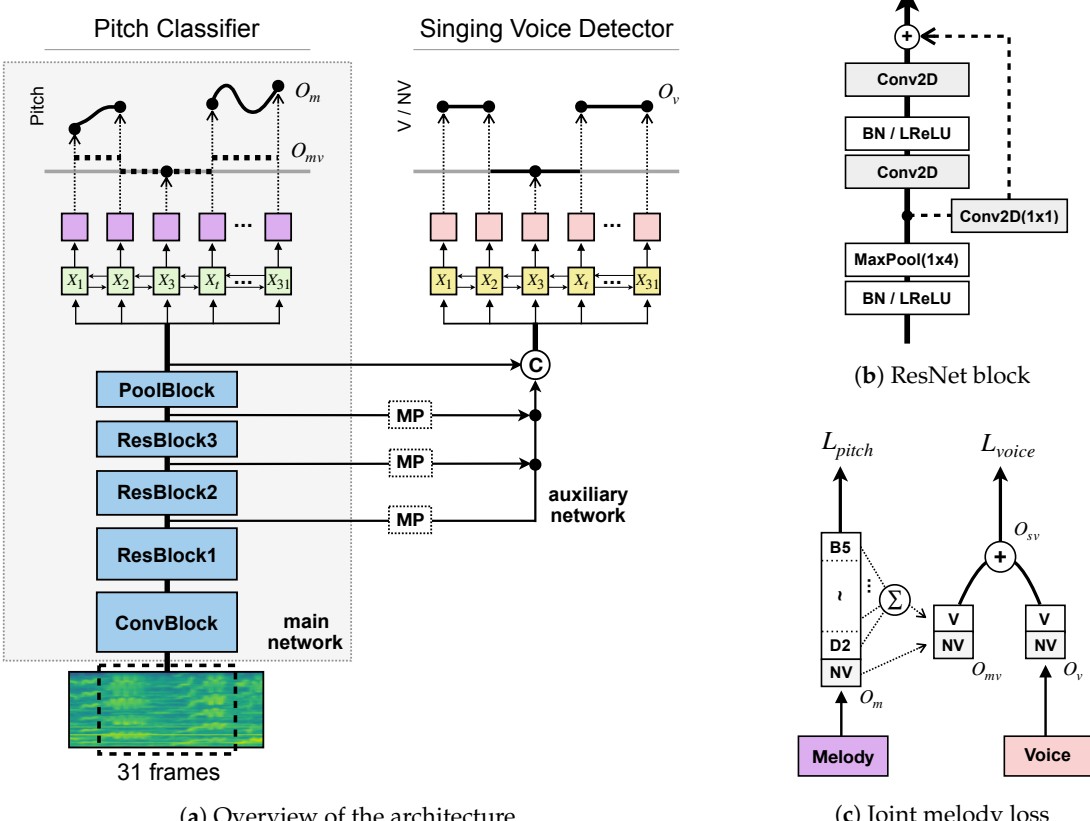

(**a**) Overview of the architecture  (**b**) ResNet block  (**c**) Joint melody loss

**Figure 1.** Overview of the architecture and joint melody loss (**a**) CRNN architectures of the main network and joint detection and classification (JDC) network for melody extraction. ConvBlock is organized in the order of 'Conv-BN-leaky rectified linear unit (LReLU)-Conv', and PoolBlock is in the order of 'BN-LReLU-MaxPool'. We denote max-pooling and concatenation as "*MP*" and "*C*", respectively. The max-pooling is applied only to the frequency axis while preserving the time-wise dimensionality. (**b**) The diagram of the ResNet block (**c**) The block diagram of joint melody loss. $o_m$ is the softmax output of the melody from the main network. $o_{mv}$ and $o_v$ are the softmax output of the singing voice activity from the main network and auxiliary network, respectively. "V" and "NV" indicate voice and non-voice, respectively.

**Table 1.** Model configurations of the main and joint networks. ConvBlock and ResBlock have two convolutional layers; $[n \times n, k]$ denotes a convolutional operator of $n$ filters and a kernel size of $k$.

|  | Components | | Output Size | |
| :---: | :---: | :---: | :---: | :---: |
|  | *Main* | *Main + AUX* | *Main* | *Main + AUX* |
| Input | | - | $31 \times 513$ | |
| Conv block | $[3 \times 3, 64] \times 2$ | | $31 \times 513, 64$ | |
| ResNet Block 1 | $[3 \times 3, 128] \times 2$ | | $31 \times 128, 128$ | |
| ResNet Block 2 | $[3 \times 3, 192] \times 2$ | | $31 \times 32, 192$ | |
| ResNet Block 3 | $[3 \times 3, 256] \times 2$ | | $31 \times 8, 256$ | |
| Pool block | | - | $31 \times 2, 256$ | |
| Bi-LSTM | 256 | 256 + 32 | $31 \times 512$ | $31 \times (512 + 64)$ |
| FC | 722 | 722 + 2 | $31 \times 722$ | $31 \times (722 + 2)$ |

ResBlock is a variation of ConvBlock that has an additional BN/LReLU, a max-pooling (MaxPool) layer with a pooling size of four, and a skip connection, inspired by ResNet of a full pre-activation version [31]. The max-pooling was only conducted in the frequency axis throughout all blocks; therefore, it preserved the input size (31 frames) in the time axis. The skip connection had an $1 \times 1$ convolution layer to match the dimensionality between two feature maps. PoolBlock was another module that consists of BN, LReLU, and MaxPool. The dropout rate of 50% was added in the end of the PoolBlock to alleviate overfitting. Finally, the Bi-LSTM layer took 31 frames of features ($2 \times 256$) from the convolution blocks (note that the max-pooling preserved the input size) and predicted pitch labels via the softmax function in a sequence-to-sequence manner.

We experimented with different kernel sizes of 1D or 2D convolution, but 2D-convolution with $3 \times 3$ filters was the most effective for melody extraction. Veit [32] showed that residual networks can be viewed as a collection of many paths. The skip connection allowed the output of each layer to be input into all subsequent blocks connected, and it made a residual network ensemble system. In our experiments, ResNet model achieved better results compared to the VGGNet [33] model. We used a total of three ResBlocks in this model. Reducing the number of ResBlocks resulted in lower performance. On the other hand, when we increased the number of ResBlocks, we could not see any noticeable performance improvement. This shows that very high-level features are not required to predict the pitch of the frame in the spectrogram.

The pitch labels ranged from D2 (73.416 Hz) to B5 (987.77 Hz) with a resolution of 1/16 semitone (i.e., 6.25 cents). Furthermore, "non-voice" (or "zero-pitch") was added to the pitch labels. This special label was active when the singing voice was not present. Therefore, the total number of labels (the size of the output layer) became 722. The main network used spectrograms as input data. Specifically, we merged audio files into a mono channel and down-sampled those to below 8 kHz, in which the majority of the singing voice spectrum is distributed. We used a 1024-point Hann window and a hop size of 80 samples (10 ms) to compute the spectrogram and compressed the magnitude in a log scale. Finally, we used 513 bins from 0 Hz–4000 Hz and 31 consecutive frames as the input of the main network.

2.1.2. Loss Function

We quantized the continuous scale of the pitch range into a discrete set of values to form the output layer in a classification setting. They are often represented as a one-hot vector to incorporate it into the categorical cross entropy (CE) loss function [15,16]. One problem of the loss function is that, unless the predicted pitch is close enough to the ground-truth pitch within the quantization size (6.25 cents in our case), it is regarded as the "wrong class". In order to mitigate excessive loss for neighboring pitches of the ground truth, a Gaussian-blurred version of one-hot vector was proposed [17,34]. We also adopted this

version, so the loss function of the main network ($L_{pitch}$) was defined between the prediction $\hat{\mathbf{y}}$ and the Gaussian-blurred labels $\mathbf{y}_g$ as below:

$$L_{pitch} = CE(\mathbf{y}_g, \hat{\mathbf{y}}) \tag{1}$$

$$y_g(i) = \begin{cases} exp(-\frac{(c_i - c_{true})^2}{2\sigma_g^2}) & \text{if } c_{true} \neq 0 \text{ and } |c_i - c_{true}| \leq M, \\ 0 & \text{otherwise,} \end{cases} \tag{2}$$

where $CE(\mathbf{y}_g, \hat{\mathbf{y}})$ is the cross-entropy loss for the pitch prediction. $c_{true}$ is the constant index of the true pitch, and $c_i$ is a variable index. $M$ determines the number of non-zero elements. We set $M$ to three and $\sigma_g$ to one in our experiment.

## 2.2. Joint Detection and Classification Network

### 2.2.1. Architecture

The output layer of the main network was formed with pitch labels and a special non-voice label. These labels were handled as a set of classes in the same level. Although voice detection and pitch estimation in the melody extraction task have a close relationship, they require different levels of abstraction, as mentioned in Section 1. In pitch estimation, the network predicts a continuously-varying value of pitch at each frame, although it uses contextual information from neighboring frames. In voice detection, the network predicts the sustained binary status of voice from textures that can be obtained from a wider context, such as vibrato or formant modulation. Therefore, simply adding a non-voice label to the target pitch labels has limitations in extracting the characteristics of voice activity.

In order to address the discrepancy between the pitch estimation and voice detection, we propose a joint detection and classification (JDC) network. We set up the JDC network so that the two tasks shared the modules. Instead of building it with new modules, we maintained the existing main network and added a branch for dedicated singing voice detection. As shown in Figure 1a, the JDC network shared ConvBlock, ResBlock, and PoolBlock in the bottom, but had a separate Bi-LSTM module for each task. In particular, the voice detection task took the combined features of the shared modules from the main network. The use of multi-level features stemmed from the idea that voice detection may require observing diverse textures in different levels of abstraction. The outputs of ResBlock were max-pooled to match the output size, and then, they were concatenated. The Bi-LSTM layer predicted the probabilities that there was a singing voice from the concatenated features in a sequence-to-sequence manner via the softmax function. We call this an auxiliary network. The features from convolutional blocks were learned jointly using the main and auxiliary network, and the loss function was combined with that derived from the main network to form the final loss function. The detail is described in the next section.

### 2.2.2. Joint Loss Function

The JDC model was optimized by minimizing a joint melody loss that combines the two loss functions from the main network and the auxiliary network, respectively, as illustrated in Figure 1c. We detected the singing voice using both networks. That is, we summed the 721 pitch predictions in the output of the main network, $o_m$, and converted them to a single "voice" prediction. This resulted in the voice detection output from the main network, $o_{mv}$. We then added this to the output of the auxiliary network, $o_v$, to make a decision for voice detection as below:

$$o_{sv} = o_{mv} + o_v \tag{3}$$

Then, the loss function for voice detection was defined as the cross-entropy between the sum of the voice output and the ground truth $\mathbf{v}_{gt}$:

$$L_{voice} = CE(softmax(o_{sv}), \mathbf{v}_{gt}) \tag{4}$$

Finally, the joint melody loss function was given by combining the loss function for voice detection ($L_{voice}$) and the loss function for pitch estimation ($L_{pitch}$):

$$L_{joint} = L_{pitch} + \alpha L_{voice} \tag{5}$$

where $\alpha$ is a balancing weight, and we used $\alpha = 0.5$ in our experiment (In our initial experiment, we tried three different values of $\alpha$ (0.1, 0.5, and 1) and achieved the best overall accuracy with 0.5 on the test datasets. Then, we fixed $\alpha = 0.5$ in the rest of the experiments. This might not be optimal, and selecting an optimal value could improve the result.).

## 3. Experiments

### 3.1. Datasets

#### 3.1.1. Train Datasets

We used the following three datasets to train the models. The songs were carefully selected so that genres and singer genders were evenly distributed for each split and songs from the same singer were not included in the other splits.

- **RWC** [35]: 80 Japanese popular songs and 20 American popular songs with singing voice melody annotations. We divided the dataset into three splits: 70 songs for training, 15 songs for validation, and the remaining 15 songs for testing.
- **MedleyDB** [36]: 122 songs with a variety of musical genres. Among them, we chose 61 songs that are dominated by vocal melody. We divided the dataset into three splits: 37 songs for training, 10 songs for validation and 12 songs for testing. For a comparison of results under the same conditions, we selected the training sets according to [18].
- **iKala** [37]: 262 Chinese songs clips of 30 s performed by six professional singers. We divided the dataset into two splits: 235 songs for training and 27 songs for validation.

We augmented the training sets by conducting pitch-shift on the original audio files by $\pm1, 2$ semitones to obtain more generalized models. Pitch-shifting has proven to be an effective way to increase data and improve results for singing voice activity detection [38], as well as melody extraction [15]. To this end, we used an algorithm based on a phase vocoder that conducts pitch-shifting independent of time-stretching [39].

#### 3.1.2. Test Datasets

Four datasets (ADC04, MIREX05 (http://labrosa.ee.columbia.edu/projects/melody/), MedleyDB, and RWC) to evaluate the performance of melody extraction were used. Non-vocal audio clips were excluded to focus on vocal melody extraction. To compare the voice detection performance, our models were also evaluated using Jamendo [40], a public dataset that is mainly used for singing voice detection.

- **ADC04**: 20 excerpts of 20 s that contain pop, jazz, and opera songs, as well as synthesized singing and audio from MIDI files. Jazz and MIDI songs were excluded from the evaluation.

- **MIREX05**: 13 excerpts that contain rock, R&B, pop, and jazz songs, as well as audio generated from a MIDI file. We used 12 songs out of a total of 20, excluding jazz and MIDI files for evaluation.
- **MedleyDB**: 12 songs not included in the training set. For a comparison of results under the same conditions, we selected the test sets according to [18].
- **RWC**: 15 songs not included in the training set. This was used internally to evaluate the performance of the proposed models.
- **Jamendo**: 93 songs designed for the evaluation of singing voice detection. Among them, only 16 songs designated as a test set to measure the performance of singing voice detection were used.

### 3.2. Evaluation

We evaluated the proposed method in terms of five metrics, including overall accuracy (OA), raw pitch accuracy (RPA), raw chroma accuracy (RCA), voicing detection rate (VR), and voicing false alarm rate (VFA). The measures are defined as below:

$$\text{RPA, RCA} = \frac{\#\{\text{voice frames for which (pitches, chromas) are predicted correctly}\}}{\#\text{ of voiced frames}} \tag{6}$$

$$\text{OA} = \frac{\#\{\text{frames for which pitches and voicing are predicted correctly}\}}{\#\text{ of all frames}} \tag{7}$$

$$\text{VR} = \frac{\#\{\text{voiced frames for which voicing are predicted correctly}\}}{\#\text{ of voiced frames}} \tag{8}$$

$$\text{VFA} = \frac{\#\{\text{frames that are predicted as voiced, but not actually voiced}\}}{\#\text{ of voiced frames}} \tag{9}$$

The evaluation consists of two main parts: voice detection determining whether the voice is included in a particular time frame (VR and VFA) and pitch estimation determining the melody pitch for each time frame (RPA, RCA). OA is the combined accuracy of pitch estimation and voice detection. We converted the quantized pitch labels in 6.25 cents (1/16 semitone) to frequency scales (Hz) to compare them with the ground truth.

$$f = 2^{(m-69)/12} \times 440 \text{ (Hz)} \tag{10}$$

where $m$ is the MIDI note number estimated by the main network for melody extraction. The pitch at each frame was considered correct if the difference between the estimated pitch and the ground-truth pitch was within a tolerance of $\pm 50$ cents. We computed them using *mir-eval*, a Python library designed for objective evaluation in MIR tasks [41].

### 3.3. Training Detail

We randomly initialized the network parameters using He uniform initialization [42] and trained them with the Adam optimizer. We repeated it over all the training data up to 45 epochs. The initial learning rate was set to 0.002. If the validation loss did not decrease within three epochs, the learning rate was reset to 80% of the previous value. Furthermore, if the validation loss did not decrease during seven epochs, the training stopped. For fast computing, we ran the code using Keras [43], a deep learning library in Python, on a computer with two GPUs.

### 3.4. Ablation Study

We conducted an ablation study to verify the effectiveness of the proposed model. We experimented with the following settings to investigate the model. The architectures of each model used in the experiments are illustrated in Figure 2.

- **The effect of the auxiliary network**: The proposed JDC network model was compared to the main network only (without the auxiliary network). They are denoted by $JDC_S$ and *Main*, respectively.
- **The effect of the combined voice detection in calculating the loss function**: The proposed model used the sum of the outputs from both the main and auxiliary networks (Equation (3)) in calculating the voice loss function. This was compared to the case where only the output of auxiliary network, $o_v$, was used in calculating the loss function, which is denoted by $JDC_A$.
- **The effect of the combined voice detection in predicting singing voice**: The JDC network can detect singing voice with three possibilities in the test phase: $o_{mv}$ from the main network, $o_v$ from the auxiliary network, and $o_{sv}$ the sum of the two outputs. We compared the performance of the three melody extraction outputs and evaluated them for each of the two loss functions above. As a result, we had a total of six outputs, which are denoted by $JDC_S(o_{mv})$, $JDC_S(o_v)$, $JDC_S(o_{sv})$, $JDC_A(o_{mv})$, $JDC_A(o_v)$, and $JDC_A(o_{sv})$.

To demonstrate the effectiveness of the JDC network, we also examined the performances of the model $Main(AUX)$ and $Main(SVD)$. In both models, the networks for pitch estimation and singing voice detector (SVD) were trained separately. Both networks for melody extraction were identical to *Main*, but only the labels corresponding to the pitches were used as the target labels. The architecture of SVD for $Main(AUX)$ was identical to the auxiliary network. Following [38], we implemented the SVD for $Main(SVD)$. For fair comparison, the same training datasets were used in all training phases.

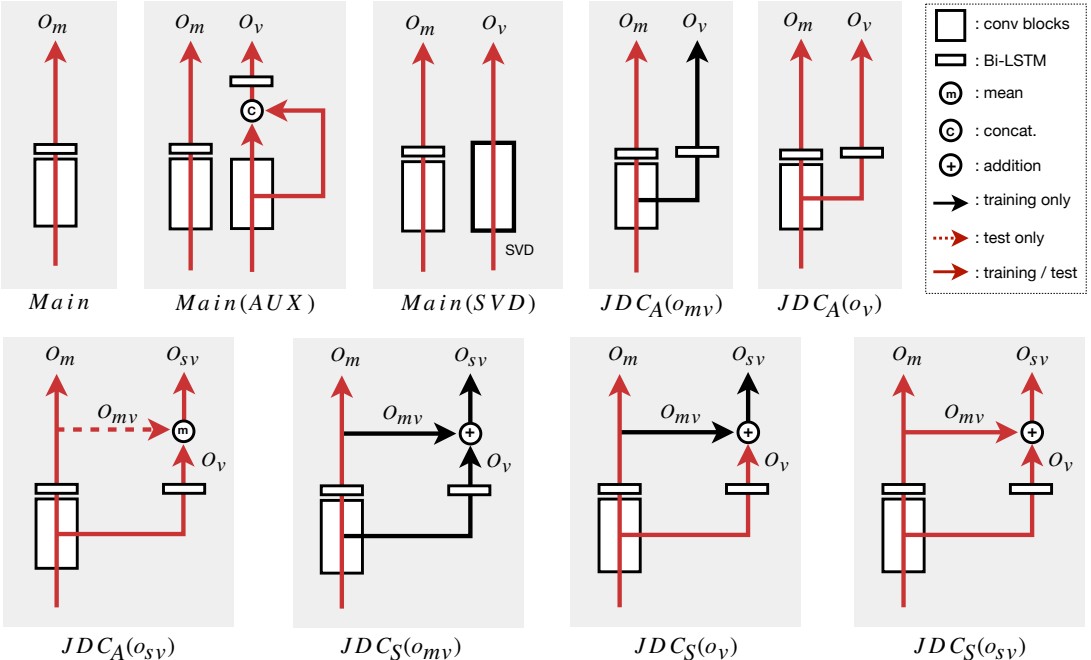

**Figure 2.** Architectures of melody extraction used for performance comparison in this paper. In the model name, the subscript refers to the type of JDC network, and parentheses refer to the source of voice detection output. The black and red solid arrows indicate the paths used for training and training/testing, respectively. The red dotted arrows indicate the path used for the test, although it was not used for training.

The models and voice detection outputs on the four test datasets were evaluated for melody extraction as mentioned in Section 3.1.2. We carried out five runs of training with different initializations and analyzed the results across different metrics of melody extraction evaluation and different test sets, respectively. To further validate the effectiveness of the models, a *t*-test between the *Main* and each JDC network to

compute the statistical significance on the results (the results were normally distributed (Shapiro–Wilk test: $p > 0.05$)) of the five trials was performed.

In addition, we examined their singing voice detection capability using the Jamendo dataset, which was dedicated to the voice detection task (there was no annotation on the pitch of the melody). Because this was for the voice detection task, we report only VR and VFA. In addition to the models and voice detection outputs in the ablation study, we also compared out best model to other state-of-the-art singing voice detection algorithms [38,44,45].

## 4. Results and Discussion

### 4.1. Comparison of Melody Extraction Performance

Figure 3a shows the results of the five melody extraction evaluation metrics for the compared models and outputs. In general, $JDC$ networks were superior to *Main* in terms of OA, and among the $JDC$ networks, $JDC_S$ networks that used the sum of the two outputs in the loss function were more accurate than $JDC_A$ that used only the output of the auxiliary network.

Both RPA and RCA increased significantly in all $JDC$ networks, especially $JDC_S(o_v)$ and $JDC_S(o_{sv})$. This is mainly attributed to the increase in VR. That is, the $JDC$ networks detected the activity of singing voice more responsively, having fewer missing errors. The average RPA and RCA of *Main* were 76.1% and 78.1%, respectively, while those of $JDC_S(o_v)$ were 84.7% and 86.0%, respectively ($p$-value < 0.01). However, both VR and VFA were high due to their aggressiveness, and this led to degradation in OA. On the other hand, $JDC_S(o_{mv})$ predicted the voice activity more reliably by significantly reducing VFA. The average VFA of $JDC_S(o_{sv})$ was 17.7%, but that of $JDC_S(o_{mv})$ was 9.0%. As a result, $JDC_S(o_{mv})$ achieved the highest average OA (85.7%, $p$-value < 0.01), outperforming the two networks.

This result indicates that the voice detection output of the main network was more conservative than the output of the auxiliary network. This is true because the main network had more classes (i.e., pitch labels) with which to compete. However, comparing $JDC_S(o_{mv})$ to $JDC_A(o_{mv})$, the main network in $JDC_S(o_{mv})$ became more sensitive to voice activity due to the influence of the auxiliary network. This reveals that combining $o_{mv}$ with $o_v$ in calculating the voice detection loss function (Equation (4)) contributed to driving more tightly-coupled classification and detection, thereby improving the performance of melody extraction.

The overall performance of *Main(AUX)* was generally higher than that of *Main*, but it did not outperform $JDC_S(o_{mv})$. The average OA of *Main(SVD)* was comparable to *Main*, and the performance was lower than that of *Main(AUX)*. Experimental results also showed that the deviations of RPA and RCA of the proposed models were high, except for *Main(AUX)* and *Main(SVD)*. Since the proposed models were trained for both pitch estimation and voice detection at different levels of abstraction, they were sensitive to initialization.

Figure 3b shows the results of overall accuracy (OA) on the four test sets for the compared models and outputs. The performance gap varied by up to 10% depending on the dataset, indicating that the models were affected by the characteristic that each test set had (e.g., genre). Again, we see that the performances of the JDC networks were generally superior to that of *Main* for all test datasets.

Comparing $JDC_A$ to $JDC_S$ in each of the three cases ($o_{mv}$, $o_v$, and $o_{sv}$), the average of OA for three $JDC_A$ and $JDC_S$ networks were 83.5% and 84.9%, respectively. $JDC_S$ networks were generally superior to $JDC_A$ networks. The average OA of $JDC_S(o_{mv})$ was improved by 3.17% over that of *Main*. With regard to OA, a $t$-test revealed a statistical significance between *Main* and $JDC_S(o_{mv})$. The results are as follows: ADC04 (0.025), MIREX05 (0.01), MedleyDB (0.027), and RWC (0.043). $JDC_S(o_{mv})$ increased the average OA with respect to *Main* for ADC04, which is an especially challenging dataset. The average overall accuracy of $JDC_S(o_{mv})$ is 83.7%, which was 6.1% higher than that of 77.6% of *Main*.

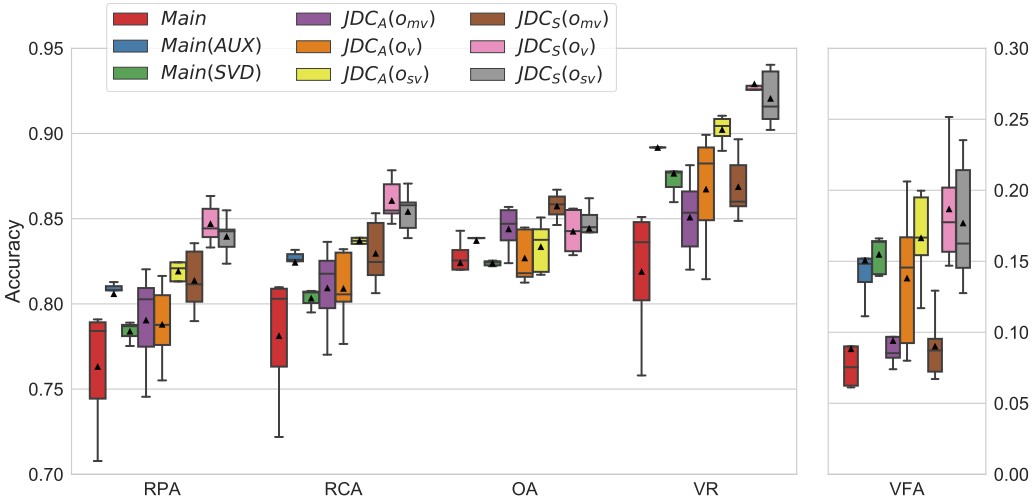

(**a**) Melody extraction accuracy for different evaluation metrics

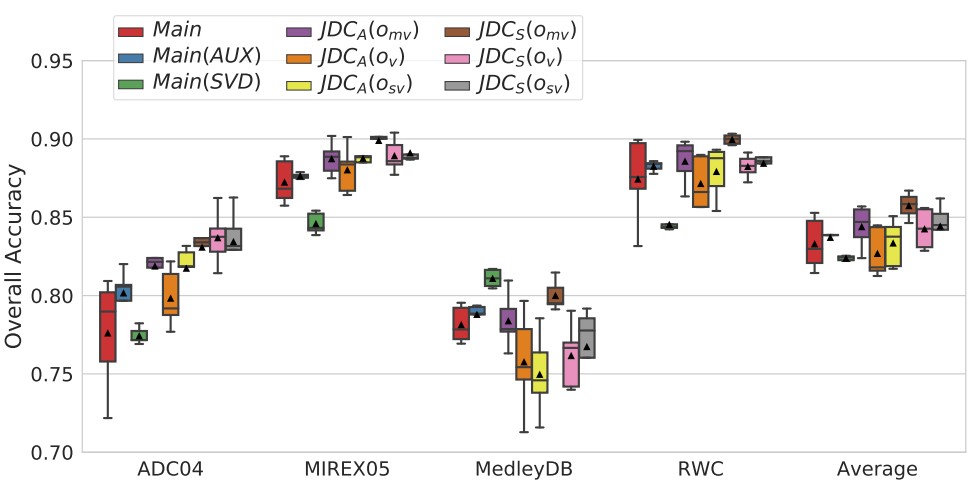

(**b**) Overall accuracy for different test datasets

**Figure 3.** Performance of melody extraction on the *Main*, *Main(AUX)*, *Main(SVD)*, and JDC networks. The reported results were averaged across five training runs with different initializations. The "average" was calculated as the average score of all songs in all datasets. The band inside the box is the median, and the black triangle indicates the mean. RPA, raw pitch accuracy; RCA, raw chroma accuracy; VR, voicing detection rate; VFA, voicing false alarm rate.

To summarize, in the training phase, the most effective models were $JDC_S$ networks that used both the main and auxiliary outputs for voice detection in the loss function. In the inference stage, the most effective output was $o_{mv}$, which used only the output of the main network. As a result, the best performance was obtained by $JDC_S(o_{mv})$. The overall performances of $Main(AUX)$ and $Main(SVD)$ were lower than the JDC networks. The JDC network had only 3.8 M parameters, while $Main(AUX)$ and $Main(SVD)$ had 7.6 M and 5.3 M parameters, respectively. It also shows that the JDC network is an efficient architecture for melody extraction.

### 4.2. Comparison of Voice Detection Performance

Figure 4 shows the average performances of singing voice detection for the *Main*, *Main*(*AUX*), *JDC_A*, and *JDC_S* networks evaluated on the four test sets. $JDC_S(O_{mv})$ achieved the best voice detection performance, leading to improved melody extraction performance. The F1 score of *Main* was 91.0%, and that of $JDC_S(O_{mv})$ was 93.3% (*p*-value < 0.05). F1 scores of other JDC networks were higher than *Main*, but there were no significant differences. For *Main*(*AUX*), *Main*(*SVD*), voice detection performance was significantly lower (the F1 scores were 87.5% and 88.9%, respectively). This seems to be due to the fact that the used training set had a higher percentage of voice segments than non-voice segments. If enough data can be used for model training, there is a possibility that the performance of SVD may be further improved.

Figure 5 displays the performances of the proposed networks evaluated on the Jamendo dataset, which is dedicated to singing voice detection and unseen in training the models. As observed in the melody extraction results, the voice detection output of the main network was more conservative. This led to a low VR and VFA. On the other hand, the *JDC* networks that had the separate singing voice detector became more responsive, having higher VR and VFA. When comparing the two families of JDC networks, *JDC_S* was more conservative than *JDC_A* as the voice loss function contained the voice output from the main network. A similar result was found among the voice detection outputs. That is, JDC with $o_{mv}$ had lower VR and VFA than JDC with $o_v$ or $o_{sv}$. While the JDC networks returned comparable results, the best performance in terms of accuracy was obtained by $JDC_S(o_{sv})$. The average of VR of $JDC_S(o_{sv})$ was 18.3% higher than that of *Main*, maintaining a low VFA of 22.6%.

In Table 2, we compare the voice detection result with other state-of-the-art algorithms. Lee et al. [46] reproduced each algorithm using the Jamendo dataset as the training data under the same conditions, and we used the results for comparison. The performance of $JDC_S(o_{sv})$ was lower; however, considering that the compared models were in fact trained with the same Jamendo dataset (by using different splits for training and testing), the result from our proposed model was highly encouraging, showing that it generalized to some extent.

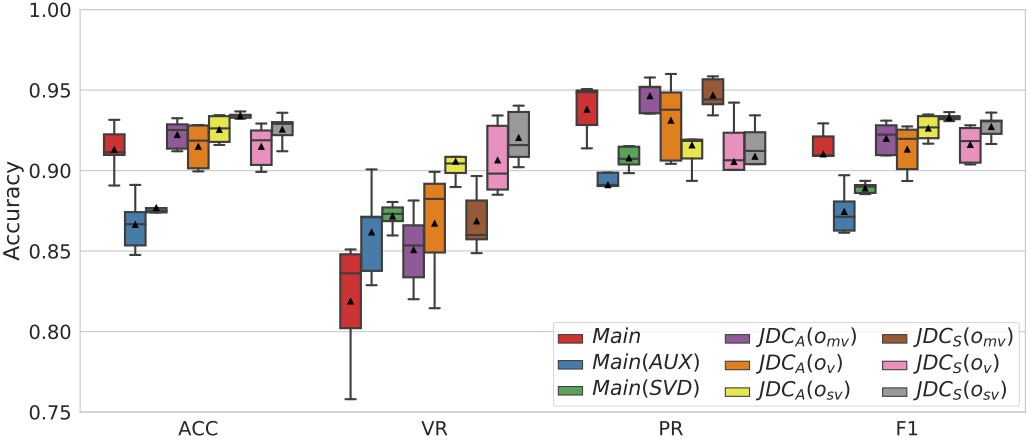

**Figure 4.** Performance of singing voice detection for different evaluation metrics. "ACC" indicates the overall accuracy of voice detection. "PR" and "F1" indicate precision and F1 score, respectively.

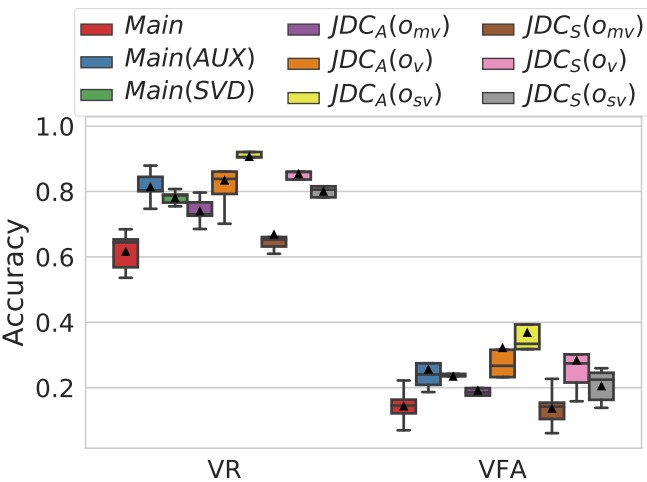

**Figure 5.** Result of singing voice detection on Jamendo.

**Table 2.** Comparison of results of existing algorithms, $Main(SVD)$, and $JDC_S(o_{sv})$ (average accuracy of 5 runs). Note that Jamendo is not included in the training set of $Main(SVD)$ and $JDC_S(o_{sv})$.

|  | Accuracy | VR | Precision | F1 Score |
|---|---|---|---|---|
| Lehner [44] | 87.9 | 91.7 | 83.8 | 87.6 |
| Schlüter [38] | 86.8 | 89.1 | 83.7 | 86.3 |
| Leglaives [45] | 87.5 | 87.2 | 86.1 | 86.6 |
| $Main(SVD)$ | 77.4 | 79.7 | 76.2 | 78.3 |
| $JDC_S(o_{sv})$ | 80.0 | 80.2 | 79.1 | 79.2 |

### 4.3. Comparison with State-of-the-Art Methods for Melody Extraction

We compared our best melody extraction model, $JDC_S(o_{mv})$, with state-of-the-art methods using deep neural networks [17,18,21,22]. For a comparison of results under the same conditions, the test sets were ADC04, MIREX05, and MedleyDB for comparing other methods as mentioned in Section 3.1.2. Table 3 lists the melody extraction performance metrics on three test datasets. The pre-trained model and code of Bittner et al. [17] are publicly available online, and the results in Table 3 were reproduced by [21] for vocal melody extraction. The results show that the proposed method had high VR and low VFA, leading to high RPA and RCA, and it outperformed the state-of-the-art methods. In addition, we confirmed that the proposed method had stable performance over all datasets compared to other state-of-the-art methods. It also showed that combining two tasks of melody extraction, i.e., pitch classification and singing voice detection, through the proposed JDC network and loss function was helpful for performance improvement.

**Table 3.** Comparison of vocal melody extraction results. The best score in each column is highlighted in bold.

| (a) ADC04 (Vocal) | | | | | |
|---|---|---|---|---|---|
| **Method** | **VR** | **VFA** | **RPA** | **RCA** | **OA** |
| Bittner et al. [17] | **92.9** | 50.5 | 77.1 | 78.8 | 70.8 |
| Su [18] | 90.1 | 41.3 | 74.7 | 75.7 | 72.4 |
| Lu and Su [21] | 73.8 | **3.0** | 71.7 | 74.8 | 74.9 |
| Hsieh et al. [22] | 91.1 | 19.2 | 84.7 | 86.2 | 83.7 |
| Proposed | 88.9 | 11.4 | **85.0** | **87.1** | **85.6** |

**Table 3.** *Cont.*

| (b) MIREX05 (Vocal) | | | | | |
| --- | --- | --- | --- | --- | --- |
| **Method** | **VR** | **VFA** | **RPA** | **RCA** | **OA** |
| Bittner et al. | **93.6** | 42.8 | 76.3 | 77.3 | 69.6 |
| Su | 95.1 | 41.1 | 83.1 | 83.5 | 74.4 |
| Lu & Su. | 87.3 | 7.9 | 82.2 | 82.9 | 85.8 |
| Hsieh et al. | 84.9 | 13.3 | 75.4 | 76.6 | 79.5 |
| Proposed | 90.9 | **2.4** | **87.0** | **87.5** | **90.7** |

| (c) MedleyDB (Vocal) | | | | | |
| --- | --- | --- | --- | --- | --- |
| **Method** | **VR** | **VFA** | **RPA** | **RCA** | **OA** |
| Bittner et al. | **88.4** | 48.7 | 72.0 | 74.8 | 66.2 |
| Su | 78.4 | 55.1 | 59.7 | 63.8 | 55.2 |
| Lu & Su | 77.9 | 22.4 | 68.3 | 70.0 | 70.0 |
| Hsieh et al. | 73.7 | **13.3** | 65.5 | 68.9 | 79.7 |
| Proposed | 80.4 | 15.6 | **74.8** | **78.2** | **80.5** |

| (d) RWC | | | | | |
| --- | --- | --- | --- | --- | --- |
| **Method** | **VR** | **VFA** | **RPA** | **RCA** | **OA** |
| Proposed | 92.4 | 5.4 | 85.4 | 86.2 | 90.0 |

*4.4. Case Study of Melody Extraction on MedleyDB*

We evaluated the models with a tolerance of one semi-tone, following the standard melody extraction evaluation rule. However, we should note that our proposed model can predict the pitch with a higher resolution (1/16 semi-tone). Figure 6a shows the spectrogram of an audio clip (top) and the corresponding melodic pitch prediction along with the ground truth (bottom). Our proposed model can track nearly continuous pitch curves, preserving natural singing styles such as pitch transition patterns or vibrato.

While the proposed model achieved improved performance in singing melody extraction, the overall accuracy was still below 90%. We found that errors occurred more frequently in particular cases. Figure 6b,c gives the examples of bad cases where VR and RPA were less than 60%. In both examples, the failures were mainly attributed to voice detection errors.

In Figure 6b, the harmonic patterns of the vocal melody were not clearly distinguished from background music because the vocal track was relatively softer than the accompanying music track. This weak vocal volume was investigated as a cause of bad singing detection in [46]. Since our melody extraction model was trained in a data-driven way, this could be addressed to some degree by augmenting the training data, for example adjusting the mixing of vocal gains (if they are in separate tracks).

In Figure 6c, a strong reverberation effect was imposed on the singing voice; thus, the harmonic patterns of the singing voice appeared even after the voice became silent. The algorithm then detected the reverberated tone as vocals and predicted the pitch from it. This case is somewhat controversial because this could be seen as a problem of the ground truth notation. When we excluded these types of heavily-processed audio clips in MedleyDB ("PortStWillow-StayEven" and "MatthewEntwistle-Lontano"), we observed a significant increase in performance (about 5% in OA on MedleyDB).

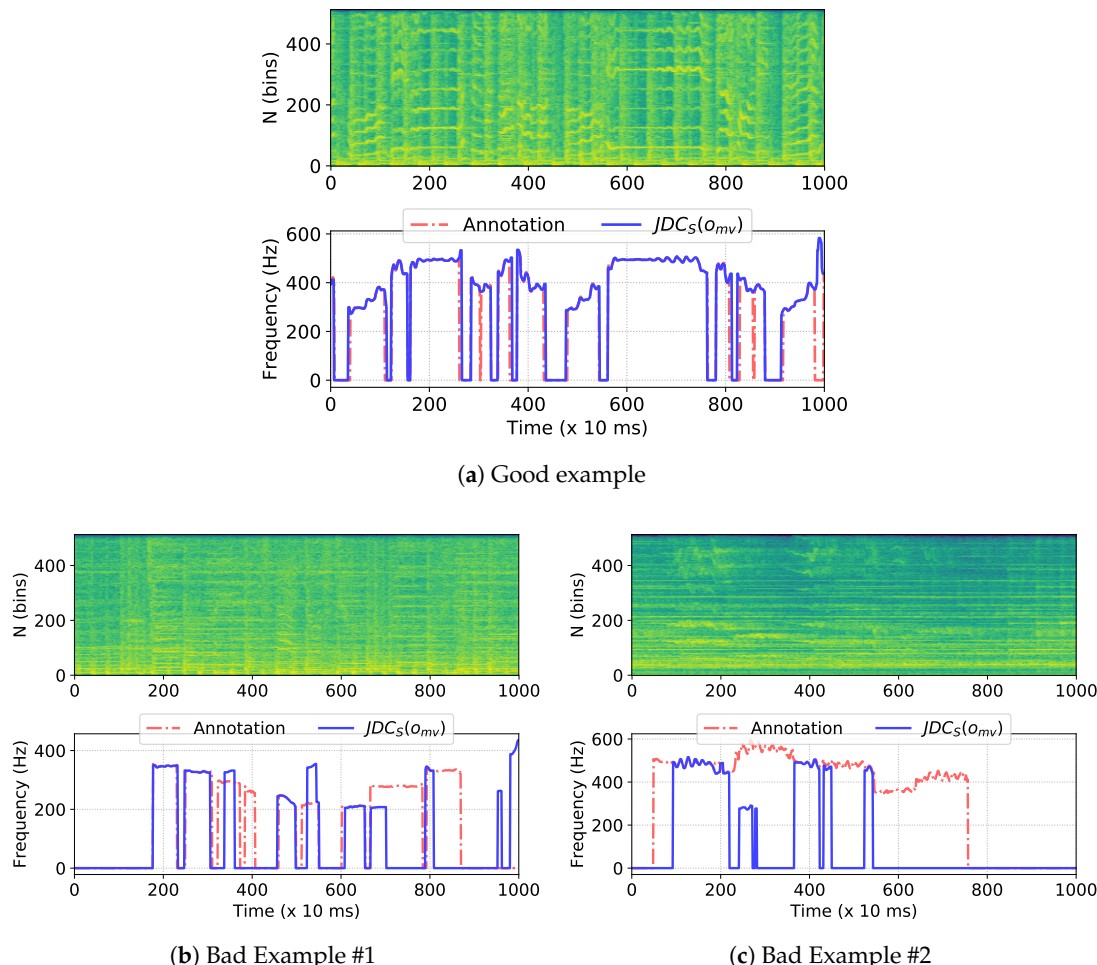

(**a**) Good example

(**b**) Bad Example #1      (**c**) Bad Example #2

**Figure 6.** Case examples of singing melody extraction from the MedleyDB dataset: (**a**) good example ("MusicDelta-Gospel") and (**b**,**c**) bad examples ("CelestialShore-DieForUs", and "MatthewEntwistle-Lontano").

## 5. Conclusions

We presented a joint detection and classification (JDC) network that performs singing voice detection and pitch estimation simultaneously. The main network uses a CRNN architecture that consists of convolutional layers with residual connections and Bi-LSTM layers. The main network is trained to classify the input spectrogram into pitch labels quantized with a high resolution or a special non-voice label. The auxiliary network is trained to detect the singing voice using only multi-level features shared with the main network. We also examined the joint melody loss function that optimizes the JDC network to combine more tightly the tasks and the three different voice detection outputs from the two networks. Through the experiment, we provided a better understanding of how the main network and auxiliary network work for voice detection. We also showed that the knowledge sharing between two networks helps perform the melody extraction task more effectively. We showed that the proposed JDC network has consistently high performance for all test datasets and outperforms previous state-of-the-art methods based on deep neural networks. Finally, we illustrated failure cases, which may provide ideas for future work to improve the melody extraction performance further.

**Author Contributions:** Formal analysis, S.K.; investigation, S.K.; methodology, S.K.; writing—original draft preparation, S.K.; writing—review and editing, J.N.; supervision, J.N.

**Funding:** This research was supported by BK21 Plus Postgraduate Organization for Content Science (or BK21 Plus Program), and Basic Science Research Program through the National Research Foundation of Korea funded by the Ministry of Science, ICT & Future Planning (2015R1C1A1A02036962).

**Conflicts of Interest:** The authors declare no conflict of interest.

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
