# Peer review of "Joint Detection and Classification of Singing Voice Melody Using Convolutional Recurrent Neural Networks"

_applsci, doi:10.3390/app9071324_

Reviewer 1 Report

The paper is correct, but it is just another paper of CRNN.

The most interesting results are Figure 4, but it doesn't explain the different between good an bad example, that it doesn't describe the features or a good or bad melody for the proposed system.

Author Response

Point1: The paper is correct, but it is just another paper of CRNN.

Response1: In 'Main melody extraction with source-filter NMF and CRNN', melody extraction is performed using CRNN. However, the main contribution of this study is not merely to apply CRNN, but to apply joint network which can improve performance of melody extraction through a joint network, which classify the pitch class and detect voice segments at the same time.

Point2: The most interesting results are Figure 4, but it doesn't explain the different between good an bad example, that it doesn't describe the features or a good or bad melody for the proposed system.

Response2: The differences between a good and bad examples are explained in Section 4.4. 

- bad example #1 : the harmonic patterns of the vocal melody are not clearly distinguished from background music.

- bad example #2 : a strong reverberation effect is imposed on the singing voice.

Reviewer 2 Report

This article presents a method to perform singing voice detection and voice pitch estimation simultaneously, in polyphonic mixtures. The authors present and evaluate different configurations, using different features and loss functions, and also compare the results against other state-of-the art approaches in the field, achieving some improvements over them.

The article provides a good background in the introduction, mentioning the relevant research in the field. The research design is appropriate, and the methods are well described. Additionally the code is publicly available which is important for reproducibility. Results are also clearly presented, but some of the conclusion lack some more rigorosity, as described in the following comments. This work has great potential to be important to the MIR community but needs a few improvements, specially regarding the scientific validity and the conclusions.

One of the aspects which could be improved is the description of the setup of the experiments and names of the methods, which is currently slightly unclear. A table would help to easily visualise the characteristics of each of the 6 methods, and would allow the reader to better follow results and the conclusions.

In the results, the authors mention significance, but it is not explicitly stated what it is meant with it. In order to have a good idea of the performance of the algorithms, and the different accuracies obtained in different songs, authors should move from plotting mean and standard deviation to draw boxplots with notches, as well as mean and median values, and additionally for some of the important conclusions also conduct some statistical significance tests.

It is also unclear how many data points have been used to compute the mean and standard deviation to plot the graphs. Is it 5 according to each of the 5 runs, taking the mean value of each metric for all songs? Or did the authors compute the std. dev of all the songs, in all the runs?

Ideally, the boxplots should show the results obtained for all songs, so that we can see the differences due to different songs.

The authors should provide some more information on why this specific architecture was selected for the estimation of pitch and voicing. E.g. why ResNet blocks? Why 3 of them? Does the performance get worse if using something simpler?

Additional experiments:

Since this work focuses on the estimation of the melody only in the case of vocal melodies, what would be the accuracy of using the best pitch estimator and the best voice detector separately, and then join the results? Is it actually important to do it jointly?

The authors should also evaluate one additional method which takes:

1) pitch estimation on one side (network trained only for predicting pitch), and

2) the estimation of voicing (voice activation) on another network trained for this specific task

Another simple additional experiment should be taking the best performing pitch estimator from the authors, but using the voice detection from some of the state of the art methods which perform best (e.g. Lehner, Schlüter, Leglaives), if their code is available. If results are better, why doing it jointly?

It would also be good to have results of the voice detection methods not only on jamendo, but also on the melody extraction datasets, for a proper comparison against the proposed methods.

Some additional comments can be found next:

In the abstract:

"in the detected segments" -> wouldn't it be better to say: "in the detected voiced segments?"

In the introduction:

"In popular music, singing voice is the main melodic" …  may be better to say:

"In popular music, singing voice is commonly / often the main melodic" … since there is also popular instrumental music, or parts of songs where the melody is played by an instrument.

“patterns of formants to deliver lyrics” → deliver lyrics sound a bit strange.. Would be worth rephrasing

“ optimal threshold value varies depending on the level” →  I would say that this depends on the pitch estimation representation too. So if would be more appropriate to say: optimal threshold value may vary depending on the level … also, there may be approaches that use a non-fix threshold (adapted to different parts of the song)

based on the the last approach. → based on the last approach.

method outperform state-of-the-art algorithms. → method outperforms state-of-the-art algorithms.

Figure 1: unclear at first sight what is CRNN and JDC. It would be better to show this in the figure itself.

“layers that sandwich a batch” → "sandwich" does not seem the most appropriate word…

2.1.2:

“LCE is the cross-entropy loss for the pitch prediction” -- > do you mean CE ? Or maybe in Eq 1, it should be LCE?

What is the value of the sigma_{g} in Eq.2?

2.2.1:

the network predicts sustained binary status of voice from wider textures such as vibrato or formant modulation. →  wider textures seems unclear in this context… consider changing the terms

"The Bi-LSTM layer predicts the voice status"  → voice status seems a bit strange… maybe voice activity status? Or simply, e.g. predicts the probability that there is singing voice in…

“We call this an auxiliary network because it uses the shared features from the main network and the loss function is combined with that derived from the main network to form the final loss function.”

This sentence is unclear… Is it not the case that the features are actually learnt jointly, using both networks?? The sentence in the article sounds as if the parameters of the main network actually learnt first, creating some features and then the second (auxiliary one)? Please clarify.

2.2.2

We detect singing voice using both the networks -- We detect singing voice using both networks

we sum the 721 pitch predictions in the output → I understand this corresponds to o_{m}, but it would be worth writing it explicitly

“We found a = 0.5 is an optimal value in our experiment.” → how was this found?? it should be specified if the ideal value was found on a validation or a test sets. Of course, ideally this should be in a validation and not test set.

“pitch detection determining the most accurate melody pitch for each time frame” → pitch detection determining the melody pitch for each time frame.

the melody pitch for each time frame (RPA, RCA, and OA). → the sentence should say that OA actually combines both pitch and voicing estimation accuracy, and maybe separate it from the rest of pitch related measures.

We conduct ablation study → We conduct an ablation study

"in calculating the loss function: the propose model" → in calculating the loss function: the proposed model

In 4.1

“Both RPA and RCA increase significantly” →  As previously introduced, proper statistical tests should be conducted, and report if there is actually a statistically significant difference.

In line 239:

improved by 2.43% over Main → clarify in which dataset

The overall accuracy of JDCS(omv) improved by 2.43% over Main, especially for MedleyDB which contains songs that are difficult to perform melody extraction. → this is not a well formed sentence… would you mean the following?:

The overall accuracy of JDCS(omv) improved by 2.43% over Main, which is especially relevant in the case of MedleyDB which is a specially challenging dataset. Maybe this sentence could be reinforced by giving here the accuracy numbers in this dataset, or by computing the percentage of improvement relative to the original accuracy, instead of in absolute value.

“This result implies that the JDC model is effective for generalizing the model” →  for generalising which other model? You mean the “Main” one? In any case, I do not think this is actually proved in the text. The authors should justify this sentence, or take it out. Why is it more generalised?

“To sum up, in the training phase, the most effective models were JDCS that uses both the main and auxiliary outputs for voice detection in the loss function.”  The authors should state what results are being compared to say that JDCs are the most effective? It does not seem that all of the JDCs obtain higher OA than all of the JDCa . The author should compare if the difference between them is statistically significant. I would suggest the comparison between JDCs and JDCa in each of the 3 cases (o_{v}, o_{sv}, o_{mv}).

In 4.2 : "JDCs generally outperform Main". The authors should say for which measure, and if the different is significative

“This leads the low VR and low VFA.” → This leads to a low VR and VFA.

The authors also use the word aggressive several times, but this may not be the ideal word in this context. Consider the use of the word lenient, or some other more appropriate word.

"In Table 2, we compare the voice detection result with those of state-of-the-art algorithms reproduced using the Jamendo dataset" : What is it meant with the word reproduced in this context?? The authors should clarify how they performed the experiments… did you use the original code of the algorithms?

4.3. Comparison with State-of-the-Arts → Comparison with the state of the art (or maybe Comparison with state-of-the-art methods)

Should Table 2 be mentioned in 4.3 (comparison against SOA) instead of in 4.2?  Or then rename 4.3 title as dealing with melody extraction only

The authors must clearly state where they got the values in Table 3 from. Were the original algorithms trained and tested on exactly the same data as the proposed methods? If they used pre-trained models, it should be clarified, as well as if they copied the results from any of the publications of the algorithms. (e.g. where do the metrics from Bittner et al. in the vocal MedleyDB come from?)

If the evaluation was performed by rerunning the algorithms on the different datasets, why is RWC not evaluated for them?

Conclusions:

shared “from” the main network →  should it be shared “with”?

Author Response

Point 1:  One of the aspects which could be improved is the description of the setup of the experiments and names of the methods, which is currently slightly unclear. A table would help to easily visualise the characteristics of each of the 6 methods, and would allow the reader to better follow results and the conclusions.

Response 1: We added Figure 2 to show the architecture of all models for clarity of explanation.

Point 2: In the results, the authors mention significance, but it is not explicitly stated what it is meant with it. In order to have a good idea of the performance of the algorithms, and the different accuracies obtained in different songs, authors should move from plotting mean and standard deviation to draw boxplots with notches, as well as mean and median values, and additionally for some of the important conclusions also conduct some statistical significance tests.

Response 2: To better demonstrate the statistical significance, we changed the bar-plot to box-plot with mean and median, and conducted t-test. We also added more details about the results in Section 4.

Point 3:  It is also unclear how many data points have been used to compute the mean and standard deviation to plot the graphs. Is it 5 according to each of the 5 runs, taking the mean value of each metric for all songs? Or did the authors compute the std. dev of all the songs, in all the runs? Ideally, the boxplots should show the results obtained for all songs, so that we can see the differences due to different songs.

Response 3

We reported the results by data set. There is a wide variety of performance differences for each song. Thus, even though there is a performance difference between models, when we report the results by song, the performance difference between 'model initialization' and 'model type' is not clearly revealed.  Also, when comparing the results of existing melody extraction, it is common to compare them by dataset.

Point 4:  The authors should provide some more information on why this specific architecture was selected for the estimation of pitch and voicing. E.g. why ResNet blocks? Why 3 of them? Does the performance get worse if using something simpler?

Response 4

We added trial/error processes and a reason why we adopted the architecture in Section 2.1.1 as follow:

"We experimented with different kernel sizes of 1D or 2D convolution, but 2D-convolution with 3x3 filters are the most effective for melody extraction. Veit shows that residual networks can be viewed as a collection of many paths. The skip connection allows the output of each layer to be input into all subsequent blocks connected and it makes a residual networks ensemble system. This can be more effective in extracting melodies that perform both pitch estimation and voice detection that require different levels of abstraction. In our experiments, ResNet model achieves better results compared to VGGNet model.

We used a total of three ResBlocks in this model. Reducing the number of ResBlocks results in lower performance. On the other hand, when we increased the number of ResBlocks, we could not see any noticeable performance improvement. This shows that very high-level features are not required to predict the pitch of the frame in the spectrogram."

Point 5: Since this work focuses on the estimation of the melody only in the case of vocal melodies, what would be the accuracy of using the best pitch estimator and the best voice detector separately, and then join the results? Is it actually important to do it jointly?

The authors should also evaluate one additional method which takes:

1) pitch estimation on one side (network trained only for predicting pitch), and

2) the estimation of voicing (voice activation) on another network trained for this specific task

Response 5:  We conducted additional experiments for Main(AUX), which use the best pitch estimator and the best voice detector separately, and then join both results. The results show that joint network is better than Main(AUX).

Point 6:  Another simple additional experiment should be taking the best performing pitch estimator from the authors, but using the voice detection from some of the state of the art methods which perform best (e.g. Lehner, Schlüter, Leglaives), if their code is available. If results are better, why doing it jointly?

Response 6:  We conducted additional experiments for Main(SVD). The singing voice detector of Main(SVD) is implemented follow Schlüter. The results show that joint network is better than Main(SVD).

Point 7: It would also be good to have results of the voice detection methods not only on jamendo, but also on the melody extraction datasets, for a proper comparison against the proposed methods.

Response 7: We added the results of the voice detection methods on 4 test sets, and reported the average accuracies of overall ACC, VR, PR, and F1.

Point 8:  In the abstract: "in the detected segments" -> wouldn't it be better to say: "in the detected voiced segments?"

Response 8: (edited) "and the other is estimating the pitch of a singing voice in the detected voiced segments."

Point 9: In the introduction:

"In popular music, singing voice is the main melodic" …  may be better to say:

"In popular music, singing voice is commonly / often the main melodic" … since there is also popular instrumental music, or parts of songs where the melody is played by an instrument.

Response 9: (edited) "In popular music, singing voice is commonly the main melodic source"

Point 10: “patterns of formants to deliver lyrics” → deliver lyrics sound a bit strange.. Would be worth rephrasing

Response 10: (edited) "they have expressive vibrato and various formant patterns unique to vocal singing."

Point 11: “ optimal threshold value varies depending on the level” →  I would say that this depends on the pitch estimation representation too. So if would be more appropriate to say: optimal threshold value may vary depending on the level … also, there may be approaches that use a non-fix threshold (adapted to different parts of the song)

Response 11: (edited) "the optimal threshold value may vary depending on the level ratio between the voice and background sound."

Point 12: based on the the last approach. → based on the last approach.

Response 12: (edited) "based on the last approach"

Point 13: method outperform state-of-the-art algorithms. → method outperforms state-of-the-art algorithms.

Response 13: (edited) "method outperforms state-of-the-art algorithms."

Point 14: Figure 1: unclear at first sight what is CRNN and JDC. It would be better to show this in the figure itself.

Response 14

- We changed the Figure1: melody contour → pitch classifier, singing voice → singing voice detector

- We have included Figure 2, which shows the architecture of melody extraction used for performance comparison.

Point 15: “layers that sandwich a batch” → "sandwich" does not seem the most appropriate word…

Response 15: (edited) ConvBlock is a module consisting of two 3×3 convolutional (Conv) layers, with a batch normalization (BN) layer, and a leaky rectified linear unit (LReLU) between them.

Point 16: 2.1.2: “LCE is the cross-entropy loss for the pitch prediction” -- > do you mean CE ? Or maybe in Eq 1, it should be LCE?

Response 16: (edited) "CE(yg,y^) is the cross-entropy loss for the pitch prediction"

Point 17: What is the value of the sigma_{g} in Eq.2?

Response 17: (edited) "sigma_{g} to 1 in our experiment."

Point 18: 2.2.1: the network predicts sustained binary status of voice from wider textures such as vibrato or formant modulation. →  wider textures seems unclear in this context… consider changing the terms

Response 18: (edited) 

"the network predicts the sustained binary status of voice from textures that can be obtained from a wider window, such as vibrato or formant modulation."

Point 19: "The Bi-LSTM layer predicts the voice status"  → voice status seems a bit strange… maybe voice activity status? Or simply, e.g. predicts the probability that there is singing voice in…

Response 19: (edited) 

"The Bi-LSTM layer predicts the probabilities that there is a singing voice from the concatenated features in a sequence-to-sequence manner via the softmax function."

Point 20:“We call this an auxiliary network because it uses the shared features from the main network and the loss function is combined with that derived from the main network to form the final loss function.”

This sentence is unclear… Is it not the case that the features are actually learnt jointly, using both networks?? The sentence in the article sounds as if the parameters of the main network actually learnt first, creating some features and then the second (auxiliary one)? Please clarify.

Response 20: (edited) 

"We call this an auxiliary network. The features from convolutional blocks are learned jointly using the main and auxiliary network, and the loss function is combined with that derived from the main network to form the final loss function."

Point 21: 2.2.2 We detect singing voice using both the networks -- We detect singing voice using both networks

Response 21: (edited) We detect the singing voice using both networks.

Point 22: we sum the 721 pitch predictions in the output → I understand this corresponds to o_{m}, but it would be worth writing it explicitly

Response 22: (edited) 

"That is, we sum the 721 pitch predictions in the output of the main network, $o_{m}$, and convert it to a single ``voice'' prediction"

Point 23: “We found a = 0.5 is an optimal value in our experiment.” → how was this found?? it should be specified if the ideal value was found on a validation or a test sets. Of course, ideally this should be in a validation and not test set.

Response 23: After several attempts, I found an appropriate alpha value, but I did not confirm the optimal value through various experiments.

(edited) "where alpha is a balancing weight and we used alpha=0.5 in our experiment."

Point 24: “pitch detection determining the most accurate melody pitch for each time frame” → pitch detection determining the melody pitch for each time frame.

Response 24: (edited) "pitch detection determining the melody pitch for each time frame."

Point 25: the melody pitch for each time frame (RPA, RCA, and OA). → the sentence should say that OA actually combines both pitch and voicing estimation accuracy, and maybe separate it from the rest of pitch related measures.

Response 25: (edited) 

"The evaluation consists of two main parts: voice detection determining whether the voice is included in a particular time frame (VR and VFA), pitch estimation determining the melody pitch for each time frame (RPA, RCA). OA is the combined accuracy of pitch estimation and voice detection."

Point 26: We conduct ablation study → We conduct an ablation study

Response 26: (edited) "We conduct an ablation study."

Point 27: "in calculating the loss function: the propose model" → in calculating the loss function: the proposed model

Response 27: (edited) "The effect of the combined voice detection in calculating the loss function"

Point 28: In 4.1 “Both RPA and RCA increase significantly” →  As previously introduced, proper statistical tests should be conducted, and report if there is actually a statistically significant difference.

Response 28: (edited)

" Both RPA and RCA increase significantly in all JDC networks, especially JDC_S(o_v) and JDC_S(o_{sv}). This is mainly attributed to the increase in VR. That is, the JDC networks detects the activity of singing voice more aggressively, having fewer missing errors. 

The average RPA and RCA of Main are 76.1% and 78.1%, respectively, while that of JDC_S(o_{v}) are 84.7% and 86.0% respectively (p-value < 0.01)."

Point 29: In line 239:

improved by 2.43% over Main → clarify in which dataset

Response 29: We changed the accuracy from OA of MeldeyDB to 'average' OA in this sentence. 

(edited) The average OA of JDC_S(o_{mv}) is improved by 3.17% than that of Main.

Point 30: The overall accuracy of JDCS(omv) improved by 2.43% over Main, especially for MedleyDB which contains songs that are difficult to perform melody extraction. → this is not a well formed sentence… would you mean the following?:

The overall accuracy of JDCS(omv) improved by 2.43% over Main, which is especially relevant in the case of MedleyDB which is a specially challenging dataset. Maybe this sentence could be reinforced by giving here the accuracy numbers in this dataset, or by computing the percentage of improvement relative to the original accuracy, instead of in absolute value.

Response 30: We changed the comparison target to ADC04, which has a larger performance difference between the two models.

(edited)  

"The average OA of JDC_S(o_{mv}) improved by over Main for ADC04, which is especially relevant in the case of ADC04 which is an especially challenging dataset. The average overall accuracy of JDC_S(o_{mv}) is 83.7%, which is 6.1% higher than that of 77.6% of Main."

Point 31: “This result implies that the JDC model is effective for generalizing the model” →  for generalising which other model? You mean the “Main” one? In any case, I do not think this is actually proved in the text. The authors should justify this sentence, or take it out. Why is it more generalised?

Response 31: we removed the sentence.

Point 32:“To sum up, in the training phase, the most effective models were JDCS that uses both the main and auxiliary outputs for voice detection in the loss function.”  The authors should state what results are being compared to say that JDCs are the most effective? It does not seem that all of the JDCs obtain higher OA than all of the JDCa . The author should compare if the difference between them is statistically significant. I would suggest the comparison between JDCs and JDCa in each of the 3 cases (o_{v}, o_{sv}, o_{mv}).

Response 32: (edited)  

"Comparing JDC_A to JDC_S in each of the three cases (o_{mv}, o_{v}, and o_{sv}), the average of OA for three JDC_A  and JDC_S networks are 83.5% and 84.9%, respectively."

Point 33: In 4.2 : "JDCs generally outperform Main". The authors should say for which measure, and if the different is significative

Response 33:  We deleted the sentence and added the results analysis.

Point 34: “This leads the low VR and low VFA.” → This leads to a low VR and VFA.

Response 34: (edited) "This leads to a low VR and VFA."

Point 35: The authors also use the word aggressive several times, but this may not be the ideal word in this context. Consider the use of the word lenient, or some other more appropriate word.

Response 35: (edited) We changed the word 'aggressive' to 'responsive'

Point 36: "In Table 2, we compare the voice detection result with those of state-of-the-art algorithms reproduced using the Jamendo dataset" : What is it meant with the word reproduced in this context?? The authors should clarify how they performed the experiments… did you use the original code of the algorithms?

Response 36: (edited) 

"In Table 2, we compare the voice detection result with other state-of-the-art algorithms. Lee et al. reproduced each algorithm using the Jamendo dataset as training data under the same conditions, and we use the results for comparison."

Point 37: 4.3. Comparison with State-of-the-Arts → Comparison with the state of the art (or maybe Comparison with state-of-the-art methods)

Response 37: (edited) "Comparison with state-of-the-art methods for melody extraction"

Point 38: Should Table 2 be mentioned in 4.3 (comparison against SOA) instead of in 4.2?  Or then rename 4.3 title as dealing with melody extraction only

Response 38: (edited) "Comparison with state-of-the-art methods for melody extraction"

Point 39: The authors must clearly state where they got the values in Table 3 from. Were the original algorithms trained and tested on exactly the same data as the proposed methods? If they used pre-trained models, it should be clarified, as well as if they copied the results from any of the publications of the algorithms. (e.g. where do the metrics from Bittner et al. in the vocal MedleyDB come from?)

Response 39: (edited) 

"The pre-trained model and code of Bittner et al. [17] are publicly available online, and the results in Table 3 were reproduced by [21] for vocal melody extraction."

Point 40: If the evaluation was performed by rerunning the algorithms on the different datasets, why is RWC not evaluated for them?

Response 40: We did not perform other methods by rerunning the algorithms, and we copied the results of them from their papers. RWC was used internally to evaluate the performance of proposed models.

Point 41: Conclusions: shared “from” the main network →  should it be shared “with”?

Response 41: (edited) 

"The auxiliary network is trained to detect the singing voice using only multi-level features shared with the main network"

Reviewer 3 Report

#9 processed -> processes

#26 formants -> formats

#44 efforts -> effort

... (I gave up)

Please proof read the paper.

2.1.1 It would help if the authors elaborate on the design principles behind their chosen architecture. If there was a trial/error process behind finalizing the architecture, the authors could briefly describe that. This will help readers who are not experienced in audio/music field to understand how deep learning architecture is adapted to this data modality.

Alternatively, if architecture in existing papers is used here, then a couple of lines of critique and reasons for adopting that architecture would be appreciated.

#137 avoid calling alpha=0.5 as 'optimal', unless it was empirically determined to be so. It is okay to use a 'reasonable ad hoc' value/ cursorily optimized value, for some hyper-parameters.

Figure #2 The use of error bars is appreciated. Of course it also leads to the question of statistical significance of the results since the relative improvement between methods in terms of accuracy is small. Furthermore, the improvement appears in some cases to be within the error bar. The authors could provide more details about this and justify that their results are statistically sound. Elaboration on random sampling, etc. for example would help.

Table #3 and Section 4.4 : A brief mention on why the proposed method outperforms the state-of-the-art methods would be appreciated towards helping the reader clearly understand how the proposed approach contrasts with related literature.

Author Response

Point 1: #9 processed -> processes

#26 formants -> formats

#44 efforts -> effort

... (I gave up)

Please proof read the paper.

Response 1: We asked the native speaker to correct the wrong English expression and the revision was completed.

Point 2 : 2.1.1 It would help if the authors elaborate on the design principles behind their chosen architecture. If there was a trial/error process behind finalizing the architecture, the authors could briefly describe that. This will help readers who are not experienced in audio/music field to understand how deep learning architecture is adapted to this data modality.

Alternatively, if architecture in existing papers is used here, then a couple of lines of critique and reasons for adopting that architecture would be appreciated.

Response 2: We added trial/error processes and a reason why we adopted the architecture in Section 2.1.1 as follow:

"We experimented with different kernel sizes of 1D or 2D convolution, but 2D-convolution with 3x3 filters are the most effective for melody extraction. Veit \cite{veit2016residual} shows that residual networks can be viewed as a collection of many paths. The skip connection allows the output of each layer to be input into all subsequent blocks connected and it makes a residual networks ensemble system. This can be more effective in extracting melodies that perform both pitch estimation and voice detection that require different levels of abstraction. In our experiments, ResNet model achieves better results compared to VGGNet \cite{simonyan2014very} model.

We used a total of three ResBlocks in this model. Reducing the number of ResBlocks results in poor feature extraction, which leads to overall performance degradation. On the other hand, when we increased the number of ResBlocks, we could not see any noticeable performance improvement. This shows that a very high-level feature is not required to predict the pitch of the frame in the spectrogram."

Point 3 : #137 avoid calling alpha=0.5 as 'optimal', unless it was empirically determined to be so. It is okay to use a 'reasonable ad hoc' value/ cursorily optimized value, for some hyper-parameters.

Response 3: (edited) "where alpha is a balancing weight and we used alpha=0.5 in our experiment." 

Point 4 : Figure #2 The use of error bars is appreciated. Of course it also leads to the question of statistical significance of the results since the relative improvement between methods in terms of accuracy is small. Furthermore, the improvement appears in some cases to be within the error bar. The authors could provide more details about this and justify that their results are statistically sound. Elaboration on random sampling, etc. for example would help.

Response 4: To better demonstrate the statistical significance, we changed the bar-plot to box-plot with mean and median, and conducted t-test. We also added more details about the results in Section 4.

Point 5 : Table #3 and Section 4.4 : A brief mention on why the proposed method outperforms the state-of-the-art methods would be appreciated towards helping the reader clearly understand how the proposed approach contrasts with related literature.

Response 5

We have added a brief mention of the outstanding results of the proposed model as follow:

"It also shows that combining two tasks of melody extraction, i.e. pitch classification and singing voice detection, through the proposed JDC network and loss function are helpful for performance improvement."

Round  2

Reviewer 2 Report

The authors have addressed the main issues described in the previous review, and they have now a higher quality publication, with still some minor issues:

With regard to previously raised points:

Response 23: After several attempts, I found an appropriate alpha value, but I did not confirm the optimal value through various experiments.

(edited) "where alpha is a balancing weight and we used alpha=0.5 in our experiment."

Still it should be explained in the article, more concretely how it was chosen… after several attempts, looking at which metric? Or if there was no optimisation, could results potentially be improved by properly selecting this value?

Response 29: We changed the accuracy from OA of MeldeyDB to 'average' OA in this sentence.

(edited) The average OA of JDC_S(o_{mv}) is improved by 3.17% than that of Main.

Clarify if the averages are calculated as the average score of all datasets, or the average score of all songs in all datasets (which would mean that larger test datasets have more influence).

Response 30: We changed the comparison target to ADC04, which has a larger performance difference between the two models.

(edited)  

"The average OA of JDC_S(o_{mv}) improved by over Main for ADC04, which is especially relevant in the case of ADC04 which is an especially challenging dataset. The average overall accuracy of JDC_S(o_{mv}) is 83.7%, which is 6.1% higher than that of 77.6% of Main."

This sentence is still very unclear or not complete… "improved by over Main for ADC04" ?? You can try to make the sentence more simple.. E.g.  JDC_S(o_{mv}) increases the OA with respect to Main for ADC04, which is an especially challenging dataset

Point 6: Another simple additional experiment should be taking the best performing pitch estimator from the authors, but using the voice detection from some of the state of the art methods which perform best (e.g. Lehner, Schlüter, Leglaives), if their code is available. If results are better, why doing it jointly?

Response 6: We conducted additional experiments for Main(SVD). The singing voice detector of Main(SVD) is implemented follow Schlüter. The results show that joint network is better than Main(SVD).

My original comment referred to using the model that e.g. Schlüter had already trained, and simply use it to estimate singing voice presence.

However, the authors reimplemented and retrained Schlüter's algorithm, but which another set of data, for which it might not be tuned.

It seems this can be observed in Figure 5 and Table 2:

Main(SVD) (which uses Schlüter's algorithm) obtains a VR of less than 80% on Jamendo, however, in Table 2 we see that the VR results on Jamendo from [38] is around 90%, and we can not see the VFA in the table.

This suggests that using the original model from [38] might get better results on the data from this article.

To have an idea of the loss of quality in the training of Schlüter's algorithm, authors should include the results of the re-implemented/retrained algorithm on Table 2.

Additionally, we also see that the best OA results on MedleyDB (which is the most realistic and broad melody extraction dataset currently) are actually obtained when using Schlüter's algorithm for voice activity detection ( Main(SVD) ), which does not seem to clearly support the fact that a joint network is beneficial.

Some further comments:

Line 10:

" The two optimizations processed are tied" ?

Do you mean:  The two optimizations processes are tied ?

Line 99:

"This can be more effective in extracting melodies that perform both pitch estimation and voice detection"

This is an unclear / incomplete sentence… This seems as if "the melodies perform both pitch estimation and voice detection" which is surely not meant to be that way.

As a general comment, please make sure all the English is properly written and makes sense.

Author Response

Response 6

We conducted additional experiments for Main(SVD). The singing voice detector of Main(SVD) is implemented follow Schlüter. The results show that joint network is better than Main(SVD).

My original comment referred to using the model that e.g. Schlüter had already trained, and simply use it to estimate singing voice presence. However, the authors reimplemented and retrained Schlüter's algorithm, but which another set of data, for which it might not be tuned. It seems this can be observed in Figure 5 and Table 2:

Main(SVD) (which uses Schlüter's algorithm) obtains a VR of less than 80% on Jamendo, however, in Table 2 we see that the VR results on Jamendo from [38] is around 90%, and we can not see the VFA in the table.

This suggests that using the original model from [38] might get better results on the data from this article.  To have an idea of the loss of quality in the training of Schlüter's algorithm, authors should include the results of the re-implemented/retrained algorithm on Table 2.

Additionally, we also see that the best OA results on MedleyDB (which is the most realistic and broad melody extraction dataset currently) are actually obtained when using Schlüter's algorithm for voice activity detection ( Main(SVD) ), which does not seem to clearly support the fact that a joint network is beneficial.

Response 6

>>

 We could not get the Schlüter's pre-trained model, so we retrained the algorithm in the same way.

>> 

Using a lot of data for training can result in higher melody extraction results. However, we have focused on validating the model by comparing the results obtained when using the same dataset in this paper. 

Therefore, the SVD of Main(SVD) used only the datasets used to train the melody extraction to make a fair comparison of the model's performance. This is mentioned in Section 3.4, and the results is shown in Figure 5. 

On the other hand, the results of Lehner, Schlüter, and Leglaives presented in Table 2 are the results of a model in which Lee [46] used both the Jamendo dataset for training and testing.

That is, the model of Schlüter and Main(SVD) have the same structure, but the results are different because of different training data.

We added the results for Main (SVD) (which uses Schlüter's algorithm) in Table 5.

>> 

VFA is an evaluation measurement used mainly in task of melody extraction rather than task of voice detection. Since the VFA is not specified in other papers, we could not  include the VFA in Table 5.

>> 

Although MedleyDB is a broadly used dataset, I do not think it can be said that the model with the highest results only in this dataset is the best. This is because a particular dataset can not represent all of the music.

In particular, Main(SVD) has low performance for other datasets except MedleyDB. This means that the SVD of Main(SVD) is overfitted to MedleyDB.  However, the JDC network is compact and has high performance for all datasets.

Point 23

Response 23: After several attempts, I found an appropriate alpha value, but I did not confirm the optimal value through various experiments.

(edited) "where alpha is a balancing weight and we used alpha=0.5 in our experiment."

Still it should be explained in the article, more concretely how it was chosen… after several attempts, looking at which metric? Or if there was no optimisation, could results potentially be improved by properly selecting this value?

Response 23

we added footnote as follow:

"(footnote) In our initial experiment, we tried three different values of alpha (0.1, 0.5, and 1) and achieved the best overall accuracy with 0.5 on the test datasets. Then, we fixed alpha=0.5 in the rest of the experiments. This might not be optimal and selecting an optimal value could improve the result."

Point 29:

Response 29: We changed the accuracy from OA of MeldeyDB to 'average' OA in this sentence. (edited) The average OA of JDC_S(o_{mv}) is improved by 3.17% than that of Main.

Clarify if the averages are calculated as the average score of all datasets, or the average score of all songs in all datasets (which would mean that larger test datasets have more influence).

Response 29:

We add the sentence in the caption in Figure 3: "The `Average' is calculated as the average score of all songs in all data sets."

Point 30:

Response 30: We changed the comparison target to ADC04, which has a larger performance difference between the two models.

(edited)  

"The average OA of JDC_S(o_{mv}) improved by over Main for ADC04, which is especially relevant in the case of ADC04 which is an especially challenging dataset. The average overall accuracy of JDC_S(o_{mv}) is 83.7%, which is 6.1% higher than that of 77.6% of Main."

This sentence is still very unclear or not complete… "improved by over Main for ADC04" ?? You can try to make the sentence more simple.. E.g.  JDC_S(o_{mv}) increases the OA with respect to Main for ADC04, which is an especially challenging dataset

Response 30: (edited) JDC_S(o_{mv}) increases the average OA with respect to Main for ADC04, which is an especially challenging dataset.

------

Some further comments:

Point 1':

Line 10: " The two optimizations processed are tied" ? Do you mean:  The two optimizations processes are tied ?

Response 1': Yes. we changed the sentence.

Point 2':

"This can be more effective in extracting melodies that perform both pitch estimation and voice detection"

This is an unclear / incomplete sentence… This seems as if "the melodies perform both pitch estimation and voice detection" which is surely not meant to be that way.

Response 2': 

We deleted this sentence because it was considered awkward in the paragraph.